# Using the Path of Least Resistance to Explain Deep Networks

## Abstract

Integrated Gradients (IG), a widely used axiomatic path-based attribution method, assigns importance scores to input features by integrating model gradients along a straight path from a baseline to the input. While effective in some cases, we show that straight paths can lead to flawed attributions. In this paper, we identify the cause of these misattributions and propose an alternative approach that treats the input space as a Riemannian manifold, computing attributions by integrating gradients along geodesics. We call this method *Geodesic Integrated Gradients* (GIG). To approximate geodesic paths, we introduce two techniques: a $k$-Nearest Neighbours-based approach for smaller models and a Stochastic Variational Inference-based method for larger ones. Additionally, we propose a new axiom, *Strong Completeness*, extending the axioms satisfied by IG. We show that this property is desirable for attribution methods and that GIG is the only method that satisfies it. Through experiments on both synthetic and real-world data, we demonstrate that GIG outperforms existing explainability methods, including IG.

## 1 Introduction

The use of deep learning models has risen in many applications. With it, so too has the desire to understand why these models make certain predictions. These models are often referred to as "opaque", as it is difficult to discern the reasoning behind their predictions Marcus (2018). Additionally, deep learning models can inadvertently learn and perpetuate biases found in their training data Sap et al. (2019). To create fair and trustworthy algorithms, it is essential to be able to explain a model's output Das & Rad (2020).

Some examples of the methods proposed to explain neural networks include Gradient SHAP Lundberg & Lee (2017), Integrated Gradients Sundararajan et al. (2017) and Guided Integrated Gradients Kapishnikov et al. (2021).

Significant effort has been dedicated to designing explanation methods that satisfy certain desirable axioms. This is due to the lack of ground truth for evaluating them. The axioms can ensure that the explanations are principled. One of the most successful axiomatic methods is Integrated Gradients (IG) Sundararajan et al. (2017). Consider a function $f : R^n \to R$, representing the neural network and an input vector $\mathbf{x} \in R^n$. Furthermore, consider a baseline input vector $\overline{\mathbf{x}} \in R^n$ (typically chosen such that the network gives baseline a near zero score). IG explains the network by quantifying how much of the difference $f(\mathbf{x}) - f(\overline{\mathbf{x}})$ can be attributed to the $i$th dimension of $\mathbf{x}$, $\mathbf{x}_i$.

Integrated Gradient gives attribution $IG_i$ to the $i$th dimension of the input by approximating the following path integral

$$IG_i(\mathbf{x}) = (\mathbf{x}_i - \overline{\mathbf{x}}_i) \int_0^1 \frac{\partial f(\gamma(t))}{\partial \mathbf{x}_i} dt, \tag{1}$$

where $\gamma(t) = \overline{\mathbf{x}} + t(\mathbf{x} - \overline{\mathbf{x}})$ is a straight path from the baseline to input. The claim of the creators of IG is that Eq. 1 tells us how the model got from predicting essentially nothing at $\overline{\mathbf{x}}$ to giving the prediction at

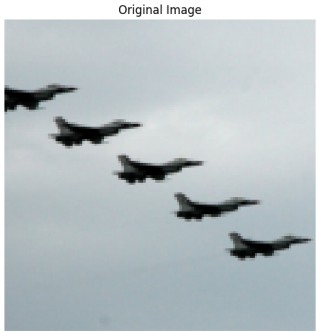 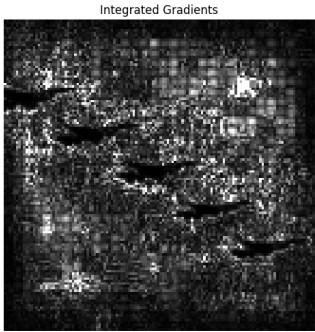 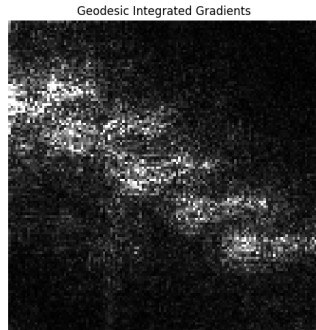

Original Image      Integrated Gradients      Geodesic Integrated Gradients

Figure 1: Comparison of attributions generated by Integrated Gradients (middle figure) and Geodesic Integrated Gradients (right figure) for image classification with a ConvNext model. Integrated Gradients follow straight paths in Euclidean space, which can result in misleading attributions. In contrast, Geodesic Integrated Gradients integrate along geodesic paths on a Riemannian manifold defined by the model, correcting misattributions caused by poor alignment with the model's gradient landscape. In both cases, the baseline is a black image. For IG, since the jets are black, apart from the artefacts created outside of the boundaries of the jets, the attribution method is misled into considering the jets unimportant for classification—despite the fact that they are the objects being classified. Geodesic IG does not suffer from this issue. Further examples of such misattributions due to black segments in images are shown in Appendix B.

$\mathbf{x}$. Considering gradients represent the rate of change of functions, the above expression should tell us how scaling each feature along the path affects the increase in the network score for the predicted class.

In this paper, we demonstrate that defining attributions along straight paths in Euclidean space can lead to flawed attributions. We examine the consequences of these misattributions through examples in computer vision, as shown in Fig. 1, along with simpler, illustrative cases in Fig. 3. To address these issues, we introduce **Geodesic Integrated Gradients**, a generalisation of IG that replaces straight paths with geodesic ones. These geodesics are defined on a Riemannian manifold, characterised by the model's input space and

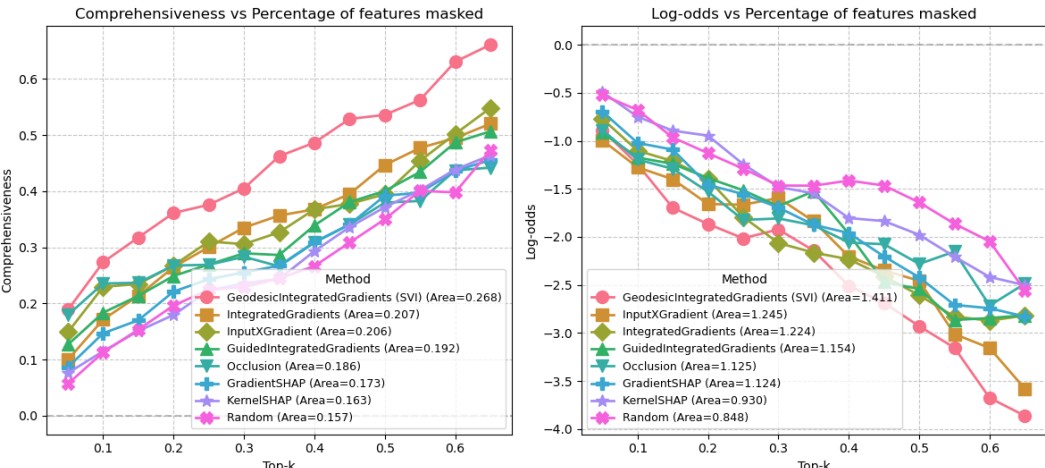

Figure 2: **Metrics Comparison.** We use a ConvNext model to classify images from the VOC dataset. The horizontal axis represents the top k% (in absolute value) of selected features. The left plot, Comprehensiveness, shows the average change in the predicted class probability compared to the original image (higher is better). The right plot displays the log-odds (lower is better). In both cases, results are summarised using AUC, where higher values indicate better performance. Geodesic IG significantly outperforms other methods in both metrics. See Section 3 for details of the experiments.

a metric tensor induced by the model's gradients. This approach mitigates the identified pitfalls while retaining all the axioms of IG. Furthermore, we introduce an additional axiom, Strong Completeness, which we argue is a desirable property for attribution methods. We prove that Geodesic Integrated Gradients is the only method that satisfies this axiom, further justifying its theoretical soundness.

Before making the case for our Geodesic Integrated Gradient, let us first show an example of an artefact that can arise from choosing straight paths, generating explanations which do not reflect the true behaviour of a model.

We highlight this issue with a simple half-moons classification task. We train a three-layer multi-layer perceptron (MLP) with ReLU activations and a cross-entropy loss to distinguish the upper moon from the lower one. The cross-entropy is decomposed into a final log-softmax activation followed by a negative log-likelihood loss, allowing us to explain probabilities. The model is trained long enough that high gradients emerge at the decision boundary, while the model remains flat elsewhere, as illustrated by the gradient contour maps in Fig. 3.

We now compute Integrated Gradients, Eq. 1, for this model on the test data. As an appropriate baseline, we choose the point $(-0.5, -0.5)$, which is a reasonable choice since the network should assign a near-zero score to it. Let us denote the feature along the vertical axis as the first component of $\mathbf{x}$, i.e., $x_1$. In Fig. 3(a) (top image), we visualise the horizontal attribution of this feature, $\text{IG}_1(\mathbf{x})$, using a colour map.

We expect in a model that is flat almost everywhere except near the decision boundary, moving slightly further from the decision boundary should not significantly change the model's score. In such a case, the attribution should remain stable for points far from the decision boundary. Yet, as shown in Fig. 3(a), Integrated Gradients significantly violates this for certain points. These are points, as seen in the figure, where the straight-line path from the baseline $(-0.5, -0.5)$ to the input passes mostly through high-gradient regions. This does not accurately reflect the model's behaviour. A similar issue arises in the vertical attribution, shown in Fig. 3(b), but in the opposite direction to the horizontal attribution.

In contrast, Geodesic Integrated Gradients (Geodesic IG), shown at the bottom of Fig. 3(a) and (b), correctly assigns equally high attributions to all points sufficiently far from the decision boundary. In Section 3.1, we detail our method and explain how it achieves the results presented in this figure.

The above artefacts further highlight an issue with over-reliance on the following axiom satisfied by IG:

**Axiom 1 (Completeness)** *Consider an input-baseline pair $\boldsymbol{x}$ and $\overline{\boldsymbol{x}}$, and a function $f$. Let $A_i(\boldsymbol{x})$ be the attribution of $\boldsymbol{x}_i$. The method satisfies Completeness if*

$$\sum_i A_i(\boldsymbol{x}) = f(\boldsymbol{x}) - f(\overline{\boldsymbol{x}}), \tag{2}$$

This axiom ensures that the total attribution equals the difference in function values between the baseline and the input. While completeness guarantees certain mathematical consistency, as the previous example illustrates, it does not account for cancellation effects between features.

In the above example, we considered a two-dimensional case where the model assigns nearly the same function value to the input points belonging to the top half-moon. Completeness then requires that for all points, the total attribution must be close to the same constant, $c$, i.e. $\sum_i A_i(\mathbf{x}) \approx c \; \forall \mathbf{x}$. Furthermore, we *know* that the horizontal feature of all points in the same half-moon have equal importance for this particular model. The same goes for the vertical features. Despite this, while IG violates the latter, it can still satisfy completeness by assigning large positive attributions to one feature and equally large negative attributions to the other—cancelling out in the sum but distorting individual attributions. This imbalance does not occur consistently across all points, further complicating interpretation.

To address this issue, we introduce a stronger axiom:

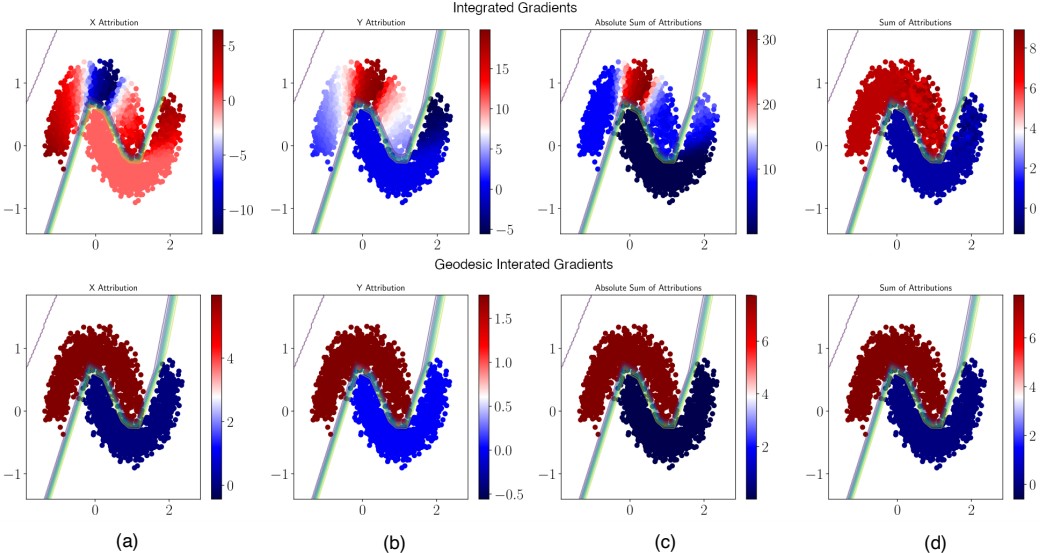

Figure 3: **Integrated Gradients (IG) attributions (top) vs. Geodesic IG (bottom).** We plot scatter plots of 10,000 samples from the half-moons dataset with noise parameter $\mathcal{N}(0, 0.15)$. An MLP model is trained for classification, and the model gradients are shown as contour maps. The model is nearly flat everywhere except at the decision boundary. Using a baseline at $(-0.5, -0.5)$, we compute IG and Geodesic IG attributions. From left to right, the colour maps display (a) feature attributions along the horizontal axis, (b) feature attributions along the vertical axis, (c) the absolute sum of attributions, $\sum_i |A_i(\mathbf{x})|$, and (d) the total sum of attributions, $\sum_i A_i(\mathbf{x})$. According to Axioms 1 and 2, the heatmaps in the last two columns should resemble those in Fig. 4. As shown, IG satisfies Axiom 1 (last column) but not Axiom 2 (penultimate column). In contrast, Geodesic IG satisfies both. Additionally, similar to Fig. 1, IG is highly sensitive to the choice of baseline due to its reliance on a straight-line path, whereas Geodesic IG mitigates this sensitivity.

**Axiom 2 (Strong Completeness)** *Consider an input-baseline pair $\boldsymbol{x}$ and $\overline{\boldsymbol{x}}$, and a continuously differentiable function $f$. Let $A_i(\boldsymbol{x})$ be the attribution of $\boldsymbol{x}_i$. The method satisfies Strong Completeness if*

$$\sum_i |A(\boldsymbol{x}_i)| = |f(\boldsymbol{x}) - f(\overline{\boldsymbol{x}})| \tag{3}$$

This stronger version of Completeness guarantees that cancellation effect cannot superficially make the attributions to add up to the difference between the function's score at the baseline versus the input points.

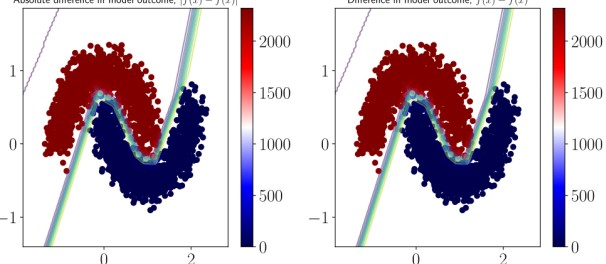

Figure 4: **Model output at the baseline vs. input points.** To assess whether our attribution methods satisfy Axioms 1 and 2 in the half-moons example, we plot the model output at the input points, subtracting the model output at the baseline. The left plot shows this difference, $f(\mathbf{x}) - f(\overline{\mathbf{x}})$, while the right plot shows the absolute difference, $|f(\mathbf{x}) - f(\overline{\mathbf{x}})|$. Comparing these plots with those in Fig. 3, we observe that Geodesic IG satisfies both axioms, whereas IG satisfies Completeness, only.

As we shall see in section 2.4.1, Geodesic Integrated Gradients is the only path-based method that satisfies this desired axiom.

In Section 2, we present two methods for approximating the geodesic path between two points on a manifold. The first method, based on $k$-nearest neighbours ($k$NN), is designed for simpler manifolds, while the second method, utilising Stochastic Variational Inference, is suited for more complex manifolds. We further demonstrate that Geodesic IG adheres to all the axioms of Integrated Gradients.

In Section 3, we demonstrate the effectiveness of the Geodesic IG method on the real-world Pascal VOC 2012 dataset Everingham et al.. Our results outperform existing methods, as we evaluate using two metrics. We preview the results of this experiment in Fig. 2.

Section 4 reviews related work, including the comparison of Geodesic IG with other methods that attempt to overcome the shortcomings of Integrated Gradients.

## 2 Method

In section 1, we gave the intuition that using geodesic paths can correct the misattributions in IG that arise from integrating along straight paths. Let us now formalise this idea.

### 2.1 Geodesic distance formulation.

Let us define a neural network as a function $f : \mathbb{R}^n \to \mathbb{R}$, where $n$ is the dimension of the input space. Let us also define $\mathbf{x}$ a point in this input space. We denote the Jacobian of $f$ at $\mathbf{x}$ as $\mathrm{J}_{\mathbf{x}}$.

Using Taylor's theorem, for a vector $\boldsymbol{\delta}$ with an infinitesimal norm: $\forall \epsilon, ||\boldsymbol{\delta}|| \leq \epsilon$, we have:

$$||f(\mathbf{x} + \boldsymbol{\delta}) - f(\mathbf{x})|| \approx ||\mathrm{J}_{\mathbf{x}}\boldsymbol{\delta}|| \approx \boldsymbol{\delta}^T \mathrm{J}_{\mathbf{x}}^T \mathrm{J}_{\mathbf{x}} \boldsymbol{\delta} \tag{4}$$

Using equation 4, we can now define a tangent space $\mathrm{T}_{\mathbf{x}}\mathrm{M}$ of all $\boldsymbol{\delta}$, equipped with a local inner product $\mathrm{G}_{\mathbf{x}}$:

$$< \boldsymbol{\delta}, \boldsymbol{\delta}' >_{\mathbf{x}} = \boldsymbol{\delta}^T \mathrm{G}_{\mathbf{x}} \boldsymbol{\delta}' = \boldsymbol{\delta}^T \mathrm{J}_{\mathbf{x}}^T \mathrm{J}_{\mathbf{x}} \boldsymbol{\delta}' \tag{5}$$

As a result, we can view the input space as a Riemannian manifold $(\mathbb{R}^n, \mathrm{G})$, where the Riemannian metric $\mathrm{G}$ is defined above. On this manifold, the length of a curve $\gamma(t) : [0, 1] \to \mathbb{R}^n$ is defined as:

$$\begin{aligned} \mathrm{L}(\gamma) &= \int_0^1 \sqrt{< \dot{\gamma}(t), \dot{\gamma}(t) >_{\gamma(t)}} dt \\ &= \int_0^1 ||\partial_t f(\gamma(t)) \times \dot{\gamma}(t)|| \, dt, \end{aligned} \tag{6}$$

where $\dot{\gamma}(t)$ is the derivative of $\gamma(t)$ with respect to $t$. The **geodesic distance**, denoted $\mathrm{L}^*$, between $\mathbf{a}$ and $\mathbf{b}$ is then defined as the minimum length among curves $\gamma$ such that $\gamma(0) = \mathbf{a}$ and $\gamma(1) = \mathbf{b}$. We also call **geodesic path** the curve $\gamma^*$ which minimises the length $\mathrm{L}$. This path can be interpreted as the shortest path between $\mathbf{a}$ and $\mathbf{b}$ in the manifold.

**Remark 1** *We can infer from Equation 6 that the geodesic path avoids as much as possible high-gradients regions. This is the main desired property of a path to be used for path-based attributions. Representing the path of least resistance, the geodesic path circumvents superficially high values of attributions.*

## 2.2 Approximation of the geodesic with $K$ Nearest Neighbours.

Computing the exact geodesic would require computing $L$ on an infinite number of paths $\gamma$, which is not possible in practice. However, several methods have been proposed to approximate this value. We draw from previous work (Yang et al., 2018; Chen et al., 2019) and present one with desirable characteristics.

First, we compute the $K$ Nearest Neighbours ($k$NN) algorithm on points between (and including) input and baseline. These points can be either sampled or generated. The geodesic distance between two neighbouring points, $\mathbf{x}_i$ and $\mathbf{x}_j$, can be approximated by a straight path $\mathbf{x}_i + t \times (\mathbf{x}_j - \mathbf{x}_i)$. We have the above approximation because for dense enough data, the euclidean distance between neighbouring points is a good approximation of the geodesic distance. This reflects the fact that a small region of a Riemannian manifold, called Riemann neighbourhood, is locally isometric to a Euclidean space[1]. So the geodesic distance between the two neighbouring points is approximated by:

$$
\begin{aligned}
\mathrm{L}_{ij}^* &= \int_0^1 ||\partial_t f(\mathbf{x}_i + t \times (\mathbf{x}_j - \mathbf{x}_i)) \times (\mathbf{x}_i - \mathbf{x}_j)||\, dt \\
&= ||\mathbf{x}_i - \mathbf{x}_j|| \int_0^1 ||\partial_t f(\mathbf{x}_i + t \times (\mathbf{x}_j - \mathbf{x}_i))||\, dt
\end{aligned}
\tag{7}
$$

Equation 7 corresponds to the original Integrated Gradients method, albeit with the norm. This integral can be approximated by a Riemannian sum similarly to Sundararajan et al. (2017):

$$
\mathrm{L}_{ij}^* \approx ||\mathbf{x}_i - \mathbf{x}_j|| \sum_{k=0}^m ||\partial f(\mathbf{x}_i + \frac{k}{m} \times (\mathbf{x}_j - \mathbf{x}_i))||
\tag{8}
$$

For input-baseline pair, $\mathbf{x}$ and $\overline{\mathbf{x}}$, we can now see the set $(\mathbf{x}, \overline{\mathbf{x}}, \mathbf{x}_i)$ as a weighted graph, with the weights being the geodesic distances between two neighbours $\mathrm{L}_{ij}^*$. To compute the geodesic path between $\mathbf{x}$ and $\overline{\mathbf{x}}$, we can use a shortest path algorithm, such as Dijkstra or $\mathrm{A}^*$ with the euclidean distance as the heuristic.

The resulting Geodesic Integrated Gradients corresponds to the sum of the gradients along this shortest path:

$$
\begin{aligned}
\text{Geodesic IG}_i(\mathbf{x}) = \\
(x_i - \overline{x}_i) \sum_{k=0}^m \int_0^1 \frac{\partial f(\mathbf{x}^k + t \times (\mathbf{x}^{k+1} - \mathbf{x}_k))}{x_i^k}\, dt
\end{aligned}
\tag{9}
$$

where $\mathbf{x}^k$ are the points along the shortest path. The integrals in Equation 9 can also be approximated with Riemannian sums.

The gradients between each pair of neighbours can also be estimated in batches to speed up the attribution computation. Moreover, several inputs' attributions can be computed together, with similar speed as IG: if we want to compute the attribution of $N$ inputs, with 10 interpolation steps and 5 nearest neighbours, the number of gradients to calculate is $10 \times 5 \times \mathrm{N} = 50\mathrm{N}$, which amounts to computing IG with 50 steps. This does not include the computation of the shortest path, which is for instance $O(\mathrm{N}^2)$ for Dijkstra algorithm. See Fig. 5 for an illustration of this method.

**Assumption of the approximation.** Here we formalise the intuition that, for a pair of neighbours, the geodesic path between them is close to the euclidean one. Notice that the derivative of the neural network $f$ is Lipschitz continuous,

---

[1]We shall further formalise this intuition later in this section.

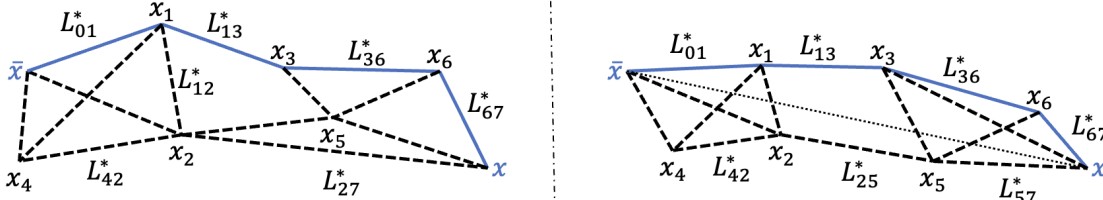

Figure 5: **Method overview.** For an input $\mathbf{x}$, a baseline $\bar{\mathbf{x}}$, and a set of points $\mathbf{x}_i$, we compute the $k$NN graph using the euclidean distance (dashed lines). For each couple $(\mathbf{x}_i, \mathbf{x}_j)$, we then compute the integrated gradients $L_{ij}^*$ using Equation 8. For clarity, not all $L_{ij}^*$ are present on the figure. 0 and 7 represent $\bar{\mathbf{x}}$ and $\mathbf{x}$ respectively. Using the resulting undirected weighted graph, we use the Dijkstra algorithm to find the shortest path between $\mathbf{x}$ and $\bar{\mathbf{x}}$ (blue continuous lines). On the left, the points $\mathbf{x}_i$ are provided while, on the right, the points are generated along the straight line between $\mathbf{x}$ and $\bar{\mathbf{x}}$ (dotted line).

$$\exists K \,\forall \mathbf{x}, \mathbf{y}, \; ||J_{\mathbf{x}} - J_{\mathbf{y}}|| \leq K \times ||\mathbf{x} - \mathbf{y}||. \tag{10}$$

Equation 10 is equivalent to the Hessian of $f$ being bounded. Under this assumption, if two points $\mathbf{x}$ and $\mathbf{y}$ are close enough, the Jacobian of one point is approximately equal to the other: if $||\mathbf{x} - \mathbf{y}|| \leq \epsilon$, then $J_{\mathbf{x}} \approx J_{\mathbf{y}}$. As a result, the length between $\mathbf{x}$ and $\mathbf{y}$, for a curve $\gamma$, is: $L(\gamma) \approx \int_\gamma ||J_{\mathbf{x}}|| \, d\mathbf{x} \approx ||J_{\mathbf{x}}|| \int_\gamma d\mathbf{x}$. Due to the triangular inequality, the shortest path $\gamma^*$ is then a straight line, and we have: $L^*(\mathbf{x}, \mathbf{y}) \approx ||J_{\mathbf{x}}|| \times ||\mathbf{x} - \mathbf{y}||$.

As a result, under this assumption, if two points are close, the geodesic path can be approximated with a straight line. Note that even though we take the path between two neighbouring points to be a straight line, we do not assume that the Jacobian of the function between the two points is constant.

**Handling disconnected graphs** An issue with the graph computed with the $k$NN algorithm is that it could be disconnected, in which case it could be impossible to compute a path between an input and a baseline. To alleviate this issue, we add so called "bridges" to the graph, as following: for each disconnected component, we add one link between them, specifically between two points of each component having the lowest euclidean distance. An illustration of this method is displayed on Figure 6.

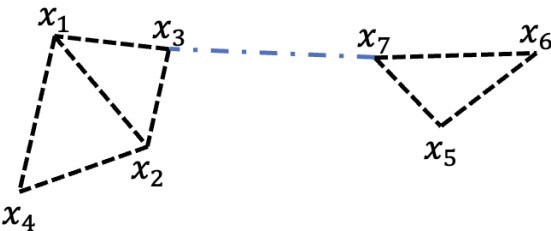

Figure 6: When the $k$NN graph is disconnected, as illustrated here, it would be impossible to compute Geodesic IG between certain points, for instance $\mathbf{x}_1$ and $\mathbf{x}_5$ here. To solve this, we add a single link between disconnected graphs, here between $\mathbf{x}_3$ and $\mathbf{x}_7$.

However, we stress that this solution is not optimal, and argue that a better way of handling this issue would be to avoid disconnected graphs in the first place. This can be done by increasing the number of neighbours k.

### 2.3 Approximation of the geodesic with energy-based sampling.

While our $k$NN-based method is effective for explaining simpler models, its applicability diminishes as model complexity increases. In such cases, a prohibitively large number of samples is required between the baseline and the input to provide accurate estimates of the geodesic path. Even with relatively large number of samples, it is not trivial where on the manifold to sample the points to adequately capture the gradient landscape. Furthermore, once the points are sampled, searching the graph for the shortest path will be computationally too intensive. For such use-cases, in this subsection, we devise an energy-base sampling method as another approximation procedure.

As noted in Remark 1, we aim to sample from the shortest paths between two points in the input space while avoiding regions of high gradients. To achieve this, we deviate from the straight line to minimise the influence of high-gradient areas. This process can be approximated as follows: we begin with a straight-line path between the two points and define a potential energy function composed of two terms: a distance term to maintain proximity to the straight line and a curvature penalty term to push away from high gradient regions. Minimising this energy function approximates the geodesic path.

Formally, the distance term is defined as $d(\mathbf{x}, \mathbf{y}) := \|\mathbf{x} - \mathbf{y}\|_2$, and the curvature term as $c(\mathbf{x}) := \|\nabla f(\mathbf{x})\|_2$ where $f$ represents the neural network. The total energy being minimised is

$$E(\gamma) = \sum_{i=1}^{n} d(\gamma_i, \gamma_i^0) - \beta c(\gamma_i), \tag{11}$$

where $\gamma$ is the path, $\gamma^0$ is the initial path, and $\beta$ controls the trade-off between distance and curvature.

With this energy function, one can use a suitable sampling method, such as Stochastic Variational Inference (SVI) or Hamiltonian Monte Carlo to sample points on the geodesic paths. Here we briefly describe the SVI optimisation, as this has a suitable balance of computational efficiency and accuracy.

SVI provides a probabilistic framework for optimising paths between input and baseline points. To achieve this, it defines a probability distribution $p(\gamma|\gamma_0)$ proportional to $exp(-E(\gamma))$, where $E(\gamma)$ is our defined potential energy. Rather than directly sampling from this complex distribution, we introduce a simpler variational distribution $q(\gamma)$ parametrised by learnable means and scales. This guide distribution takes the form of a factorised normal distribution $\prod_i N(\mu_i, \sigma_i)$ over path deviations.

The optimisation proceeds by minimising the KL divergence between $q(\gamma)$ and the true posterior through maximisation of the Evidence Lower Bound. Critically, this allows us to learn optimal parameters for $q(\gamma)$ through gradient-based optimisation. The learned means $\mu_i$ define the optimal path deviations from the initial straight-line path, while the scales $\sigma$ capture uncertainty in these deviations. This probabilistic approach naturally samples of the low-energy regions.

We apply this method in our computer vision experiments, demonstrating its efficacy in Section 3. However, for clarity, we visualise these paths on a simpler 2D half-moons example in Fig. 7. While the $k$NN method would typically be preferred for such simpler cases due to its ease of control, this example serves as an instructive illustration.

Of course, using the SVI method comes with its own challenges. For example, in this case a suitable value of $\beta$ for the potential energy, as well as learning rate for the SVI algorithm itself needs to be chose. As for any standard machine learning training, these values can be chosen using hyperparameter tuning, as we discuss in section 3.

### 2.4 Axiomatic properties

Designing an effective attribution method is challenging, partly because there are often no ground-truth explanations. If a method assigns importance to certain features in a counterintuitive way, it can be difficult to determine whether the issue lies with the model, the data, or the attribution method itself.

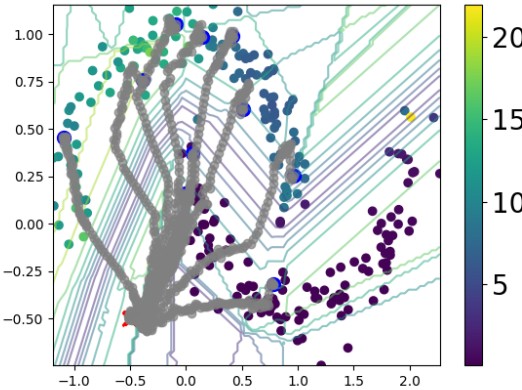

Figure 7: **Visualisation of 10 random paths.** For the simple case of half-moons with 1,000 samples, we display the sampled paths between 10 pairs of points. In low-gradient regions, the sampler favours straight lines, whereas in high-gradient regions, the paths adjust to become nearly perpendicular to the large gradient vectors, crossing these regions as quickly as possible.

To address this, some attribution methods follow an axiomatic approach, defining principles that guide their design. Integrated Gradients is one such method, satisfying several key axioms. As shown in Sundararajan et al. (2017), all path-based methods (a generalisation of IG to arbitrary paths) uphold these axioms except for Symmetry.

Additionally, as discussed in Section 1, beyond IG's axioms, we require Strong Completeness, Axiom 2. Therefore, this subsection focuses on Strong Completeness and Symmetry only.

### 2.4.1 Strong Completeness

Here we prove that Geodesic IG satisfies Axiom 2, Strong Completeness, and is the only path-based method that does so.

**Theorem 1 (Strong Completeness)** *Let $f : \mathbb{R}^n \to \mathbb{R}$ be continuously differentiable, and let $\overline{\mathbf{x}}, \mathbf{x} \in \mathbb{R}^n$. Given a smooth path*

$$\gamma : [0,1] \to \mathbb{R}^n, \quad \gamma(0) = \overline{\mathbf{x}}, \quad \gamma(1) = \mathbf{x},$$

*define its attributions as*

$$A_i^\gamma(\mathbf{x}) = \int_0^1 \frac{\partial f}{\partial x_i}(\gamma(t)) \, \dot{\gamma}_i(t) \, dt, \tag{12}$$

*and assume the Riemannian metric is given by Eq. 5. The length of a path is given by Eq. 6. Suppose that the geodesic path connecting $\overline{\mathbf{x}}$ and $\mathbf{x}$ exists. Then,*

$$\sum_{i=1}^n \left| A_i^\gamma(\mathbf{x}) \right| = \left| f(\mathbf{x}) - f(\overline{\mathbf{x}}) \right|$$

*if and only if $\gamma$ is the geodesic path.*

See Appendix A for the proof.

### 2.4.2 Symmetry preserving of Geodesic IG

The symmetry axiom is defined in the following way.

**Axiom 3 (Symmetry)** *Consider an input-baseline pair $\boldsymbol{x}$ and $\overline{\boldsymbol{x}}$, and a function $f$ that is symmetric in dimensions $i$ and $j$. If $\boldsymbol{x}_i = \boldsymbol{x}_j$ and $\overline{\boldsymbol{x}}_i = \overline{\boldsymbol{x}}_j$, then an attribution method is Symmetry-Preserving if $A_i(\boldsymbol{x}) = A_j(\boldsymbol{x})$, where $A_n(\boldsymbol{x})$ is the attribution of $\boldsymbol{x}_n$.*

(Sundararajan et al., 2017, Theorem 1) shows that IG is the only path-based attribution method that satisfies symmetry for any function. However, as noted in Kapishnikov et al. (2021), while the straight path is the only one satisfying symmetry for *any* function, for a specific function, it may be possible to find other paths that also satisfy symmetry. Below, we demonstrate that Geodesic IG satisfies symmetry for Riemannian manifolds, and thus for the neural network functions we use when sampling the paths.

Let the $i$th and $j$th dimensions of $\gamma(t)$ be $\gamma_i(t)$ and $\gamma_j(t)$ respectively and $f$ be a function differentiable almost everywhere on $t$. Furthermore, take $f$ to be symmetric with respect to $x_i$ and $x_j$. If $\gamma_i(t) = \gamma_j(t)$ for all $t \in [0,1]$, then we have

$$||\partial_t f(\gamma_i(t)) \times \dot{\gamma}_i(t)|| = ||\partial_t f(\gamma_j(t)) \times \dot{\gamma}_j(t)||, \tag{13}$$

almost everywhere on $t$. Therefore, the $i$th and $j$th components of Eq. 6 are equal. Furthermore, since Eq. 9 integrates along the path that is an approximation of Eq. 6, we have Geodesic $\text{IG}_i$ = Geodesic $\text{IG}_j$. Indeed our geodesic paths satisfy $\gamma_i(t) = \gamma_j(t)$ for all $t \in [0,1]$ on the Riemannian manifolds. To see this, let us select a baseline $\overline{\mathbf{x}}$ and $U$ a Riemann neighbourhood centred at $\overline{\mathbf{x}}$. Let us also define the geodesic path $\gamma$ such as $\gamma(0) = \overline{\mathbf{x}}$. Further, define $\mathbf{v}(t) := \gamma'(t)$, where $\gamma'$ is the derivative of $\gamma$. Then, in the local coordinates system of the neighbourhood of any point, called normal coordinates, we have $\gamma(t) = (tv_1(t), ..., tv_n(t))$. Since the function is symmetric in the $i$th and $j$th dimensions, we have $v_i$ and $v_j$ are the same everywhere. From this, we can see that $\gamma_i(t) = \gamma_j(t)$ for all $t \in [0,1]$ and therefore Geodesic IG satisfies symmetry.

## 3 Experiments

To validate our method, we performed experiments on two datasets: one is the synthetic half-moons dataset, and the other is the real-world Pascal VOC 2012 dataset.

### 3.1 Experiments on the half-moons dataset

We use the half-moons dataset provided by Scikit learn (Pedregosa et al., 2011) to generate 10,000 points with a Gaussian noise of $\mathcal{N}(0, x)$, where $x$ ranges between 0.05 and 0.65. The dataset is split into 8,000 training points and 2,000 testing ones. The model used is an MLP.

We evaluate each attribution method using an indicator of performance: the absence of artefacts that do not reflect the model's behaviour. To this end, we use purity, defined as follows.

A well-trained model should classify approximately half of the data points as "upper moon" (class 1) and the other half as "lower moon" (class 0). Such a model should consider both features of each point important for classification into class 1. Therefore, for a good attribution method, A, we expect the top 50% of points—ranked by the quantity $\widetilde{A}(\mathbf{x}) = \sum_{i=0}^{1} |A_i(\mathbf{x})|$, to be classified as 1, assuming the baseline is chosen as a point to which the network assigns a near-zero score. With this in mind, we define purity as

$$\text{Purity} = \frac{1}{N/2} \sum_{\mathbf{x},\, \widetilde{A}(\mathbf{x}) \in \text{Top 50\% of all A}} \text{argmax}(f(\mathbf{x})), \tag{14}$$

where $N$ is the number of data points. We see that this is the average value of the predicted class labels for half of the points. From the above, we infer that, for a well-trained model, we prefer an attribution method that results in the purity close to 1. In contrast, a random attribution method in this case would result in the purity score of 0.5.

In this experiment, we compare the results of attributions from Geodesic IG with methods including Integrated Gradients, GradientShap, InputXGradients (Shrikumar et al., 2016), KernelShap (Lundberg & Lee, 2017), Occlusion (Zeiler & Fergus, 2014), and Guided IG (Kapishnikov et al., 2021).

| Method | AUC-Purity ↑ |
|---|---|
| Input X Gradients | 0.328 |
| GradientShap | 0.483 |
| IG | 0.487 |
| Random | 0.299 |
| Kernel Shap | 0.480 |
| Occlusion | 0.520 |
| Guided IG | 0.361 |
| Enhanced IG | 470 |
| Geodesic IG ($k$NN) | **0.531** |
| Geodesic IG (SVI) | 0.504 |

Table 1: Evaluation of different attribution methods on a half-moons dataset with Gaussian noises with standard deviation ranging from 0.05 to 0.65. While our $k$NN-based method outperforms all other methods, we see that unlike larger examples, such as the one summarised in Table 2, our SVI example struggles to compete due to complexity of tuning hyperparameters.

For all of the methods, we use $(-0.5, -0.5)$ as a baseline. The chosen number of neighbours for the $k$NN part of both Enhanced IG and Geodesic IG is 15.

We have run our experiment with 5 different seeds and plotted the mean and standard error of the results for different noise levels in Fig. 8. We have summarised these results by reporting their area under the curve (AUC) in Table 1. We see that our $k$NN-based Geodesic IG outperforms all other methods, with the gap increasing with noise. While Occlusion comes close second, we see in Table 2 that the method does not perform well in larger examples with more complex embeddings. To provide better understanding of the comparison of our results with Enhanced IG we present more analysis on this dataset in Section 4.

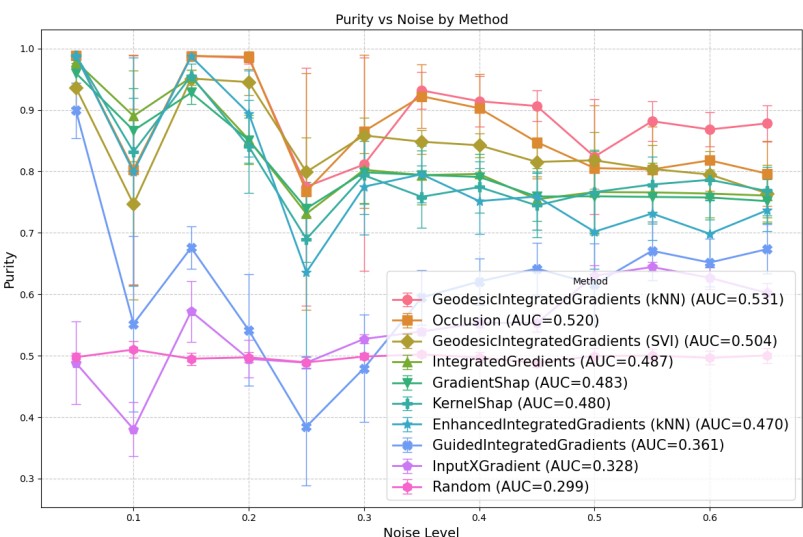

Figure 8: A comparison of different attribution methods on half-moons dataset with noise levels ranging from 0.05 to 0.65. Our $k$NN-based method outperforms all, with the gap increase with noise.

## 3.2 Experiments on the Pascal VOC 2012 dataset

To evaluate our method on a real-world dataset, we used the Pascal VOC 2012 dataset (Everingham et al.), which consists of labelled images. We trained a classification head on this dataset and integrated it with the pre-trained ConvNext model (Liu et al., 2022) from TorchVision to generate predictions for explanation.

| Method | AUC-Comp ↑ | AOC-LO ↑ |
|---|---|---|
| Input X Gradients | 0.21 | 1.28 |
| GradientShap | 0.18 | 1.15 |
| IG | 0.21 | 1.25 |
| Random | 0.16 | 0.86 |
| Kernel Shap | 0.16 | 0.94 |
| Occlusion | 0.19 | 1.16 |
| Guided IG | 0.20 | 1.18 |
| Geodesic IG (SVI) | **0.27** | **1.44** |

Table 2: Evaluation of different attribution methods on 100 randomly sampled images from the Pascal VOC test set. Fig. 2 shows the curves where these metrics are extracted from.

For this experiment, we used the same attribution methods as our half-moons experiment, except for the $k$NN-based methods (Enhanced IG and Geodesic IG ($k$NN)) since densely sampling and searching the gradient space of a large model was not practical. We applied these attribution methods to classification of 100 randomly selected images. For explainers that require a baseline, such as IG and our proposed method, a uniformly black image was used as the baseline.

To measure the performance of an attribution method, we use 2 different metrics:

- **Comprehensiveness** (DeYoung et al., 2019): We mask the top k% most important features in absolute value, and compute the average change of the predicted class probability compared with the original image. A higher score is better as it indicates masking these features results in a large change of predictions.

- **Log-odds** (Shrikumar et al., 2017): We mask the top k% most important features in absolute value, and measure the log-odds. Lower scores are better.

We evaluated these metrics for a range of top-$k$%, from 1% to 65%, as shown in Fig. 2. To summarise performance across different $k$% values, we calculated the area under the curve (AUC) for Comprehensiveness. We similarly calculated the area over the curve (AOC) for Log-odds, since the values of this metric are negative. In both cases these are the area between the metric curves and the horizontal line at $y = 0$. These results are presented in Table 2.

Additionally, Fig. 1 provides a qualitative comparison between Geodesic IG and Integrated Gradients. The results demonstrate that Geodesic IG outperforms other methods in explaining the model's behaviour on the dataset, with a particularly notable improvement in comprehensiveness. Further qualitative comparisons can be found in Appendix B.

Using Geodesic IG to explain complex deep learning models comes with some challenges. One issue is the high computational cost of sampling methods like SVI or HMC. For instance, running the aforementioned experiment on 100 images required 23 hours on an L4 GPU. While computationally expensive, this could be justified in scenarios where accurate explanations are crucial. Another challenge is that the performance of the energy-based geodesic method depends on selecting the right value for $\beta$ in Eq. 11 and tuning SVI hyperparameters, such as the learning rate. In principle, hyperparameter tuning using metrics like Comprehensiveness or Log-odds could help optimise these parameters. However, due to limited computational resources, we were unable to perform such tuning in this study, though it has the potential to significantly improve results. Lastly, the optimisation nature of these sampling methods can cause the endpoints of the paths to deviate slightly from the baseline and input points, favouring nearby points in lower gradient regions. To address this, we added a term to the potential energy to correct for these deviations and ensure more accurate alignment with the intended points. Formally, the extra term is $w \sum_{t \in \text{endpoints}} |\gamma(t) - \gamma_{\text{init}}(t)|_2$, where $w$ is endpoint weight and the first/ last 10% of the paths are counted as endpoints .

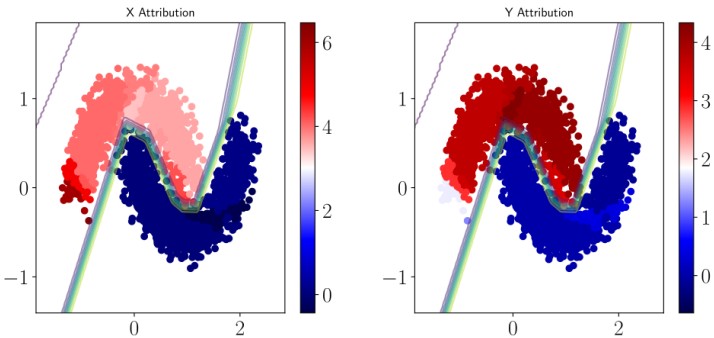

Figure 9: **Enhanced IG attributions.** Enhanced IG applies a $k$-NN algorithm, then uses Dijkstra's algorithm to find the shortest path between an input and a reference baseline, and computes gradients along this path. However, since this method is model-agnostic, it does not account for high-gradient regions. As a result, the attributions exhibit an unjustified shading that does not accurately reflect the model's true behaviour. Comparing the horizontal and vertical attributions (left and right plots, respectively), we confirm that this method does not satisfy Axiom 2, as expected. In this example, similar to those in Fig. 3, we sample 10,000 points from the half-moons dataset with noise drawn from $\mathcal{N}(0, 0.15)$.

## 4 Related Work

Approximating geodesic paths is a widely studied area of research, and many methods to do so have been developed. For a comprehensive survey on this subject, please refer to Crane et al. (2020).

The idea of using a $k$NN algorithm to avoid computing gradients on out of distribution data points has also been used in Enhanced Integrated Gradients Jha et al. (2020). However, this method creates a path which is model agnostic, as it does not necessarily avoid high gradients regions. As a result, it can lead to significant artefacts which do not reflect the model's behaviour. To support this argument, we provide an example where this method fails on the half-moons datasets. In Fig. 9, similar to the example in section 1, we see the Enhanced IG attributes different importance to the horizontal and vertical features of the half-moon data points, in a model that is flat everywhere, other than the decision boundary. Furthermore, in Fig. 9, we observer that the method violates Strong Completeness axiom, Axiom 2. This is expected, given Theorem 1.

The idea of adapting the path to avoid high gradient regions has be proposed by Kapishnikov et al. (2021), calling their method Guided Integrated Gradients. This method has a heuristic approach to finding such paths. As a result it does not guarantee to find paths of minimal accumulated gradients. In contrast, Geodesic Integrated Gradients offers a more principled approach by directly approximating the path of least resistance using a Riemannian manifold framework. As a result, as we see in Section 3, our method significantly outperforms Guided IG in terms of attribution accuracy.

## 5 Discussion

In this paper, we identified key limitations of path-based attribution methods such as Integrated Gradients (IG), particularly the artefacts that arise from ignoring the model's curvature. To address these issues, we introduced a novel path-based method, Geodesic IG, that integrates gradients along geodesic paths on a manifold defined by the model, rather than straight lines.

By avoiding regions of high gradient in the input space, Geodesic IG effectively mitigates these artefacts while preserving all the axioms established by Sundararajan et al. (2017). Additionally, we introduced a new axiom, Strong Completeness, which, when satisfied, prevents such misattributions. We proved that Geodesic IG is the only path-based method that satisfies this axiom. Through both theoretical analysis and empirical evaluation—using metrics such as Comprehensiveness and Log-Odds—we demonstrated the advantages of our approach.

To approximate geodesic paths, we proposed two methods: one based on $k$-Nearest Neighbour and another leveraging Stochastic Variational Inference. While these methods outperform existing alternatives, they also present challenges. One such challenge is computational cost, as discussed in Section 3. Another is the inherent noise in sampling-based geodesic approximations. Even though in our experiments we demonstrated noise reduction relative to the original IG, we believe further improvements can be achieved. A promising future direction is to solve the geodesic equation directly, which could reduce noise and improve accuracy. Additionally, depending on the chosen solution method, this approach may offer greater computational efficiency compared to the current reliance on SVI.

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

# A   Proof of Theorem 1

**Proof 1** *We proceed in two parts: first, we show that if $\gamma$ is the geodesic path, then the equality holds; second, we show that if the equality holds, then $\gamma$ must be the geodesic path.*

### *Sufficiency: If $\gamma$ is the geodesic, then the equality holds.*

*Let $\gamma$ be the geodesic path connecting $\overline{\mathbf{x}}$ and $\mathbf{x}$. By definition of a geodesic in the given Riemannian metric, $\gamma$ satisfies the geodesic equation:*

$$\nabla_{\dot\gamma}\dot\gamma = 0,$$

*where $\nabla$ is the Levi-Civita connection associated with the metric $\mathbf{G_x}$. Under the given metric $\mathbf{G_x} = \nabla f(\mathbf{x})\nabla f(\mathbf{x})^T$, the geodesic equation implies that $\dot\gamma(t)$ is parallel to $\nabla f(\gamma(t))$ for all $t \in [0, 1]$. That is,*

$$\dot\gamma(t) = \lambda(t)\nabla f(\gamma(t)),$$

*for some scalar function $\lambda(t)$.*

*Substituting this into the definition of $A_i^\gamma(\mathbf{x})$, Eq. 12, we obtain:*

$$A_i^\gamma(\mathbf{x}) = \int_0^1 \frac{\partial f}{\partial x_i}(\gamma(t))\dot\gamma_i(t)\, dt = \int_0^1 \frac{\partial f}{\partial x_i}(\gamma(t))\lambda(t)\frac{\partial f}{\partial x_i}(\gamma(t))\, dt.$$

*Since $\lambda(t)$ does not change sign (as $\gamma$ is a geodesic²), the integrand $\frac{\partial f}{\partial x_i}(\gamma(t))\dot\gamma_i(t)$ maintains a consistent sign across all $i$ and $t$. Therefore,*

$$\sum_{i=1}^n |A_i^\gamma(\mathbf{x})| = \left|\sum_{i=1}^n A_i^\gamma(\mathbf{x})\right|.$$

*By the Fundamental Theorem of Calculus,*

$$\sum_{i=1}^n A_i^\gamma(\mathbf{x}) = \int_0^1 \frac{d}{dt} f(\gamma(t))\, dt = f(\mathbf{x}) - f(\overline{\mathbf{x}}).$$

*Thus,*

$$\sum_{i=1}^n |A_i^\gamma(\mathbf{x})| = |f(\mathbf{x}) - f(\overline{\mathbf{x}})|.$$

### *Necessity: If the equality holds, then $\gamma$ is the geodesic.*

*Suppose $\gamma$ is a smooth path such that*

$$\sum_{i=1}^n |A_i^\gamma(\mathbf{x})| = |f(\mathbf{x}) - f(\overline{\mathbf{x}})|.$$

*By the triangle inequality,*

$$\sum_{i=1}^n |A_i^\gamma(\mathbf{x})| \geq \left|\sum_{i=1}^n A_i^\gamma(\mathbf{x})\right| = |f(\mathbf{x}) - f(\overline{\mathbf{x}})|.$$

---

²Since $\gamma$ is a geodesic, its velocity is always parallel to the gradient of $f$, so there exists a scalar function $\lambda(t)$ such that

$$\dot\gamma(t) = \lambda(t)\nabla f(\gamma(t)).$$

Then, by the chain rule,

$$\frac{d}{dt} f(\gamma(t)) = \nabla f(\gamma(t)) \cdot \dot\gamma(t) = \lambda(t)\|\nabla f(\gamma(t))\|^2.$$

Assuming $\|\nabla f(\gamma(t))\|^2 > 0$, the sign of $\frac{d}{dt} f(\gamma(t))$ is completely determined by the sign of $\lambda(t)$. Since $f(\gamma(t))$ changes monotonically (i.e., without backtracking) from $f(\gamma(0))$ to $f(\gamma(1))$, $\lambda(t)$ must maintain a constant sign along $\gamma$.

Intuitively, $\dot\gamma(t) = \lambda(t)\nabla f(\gamma(t))$ means that $\gamma$ is always moving in the direction of steepest ascent (or descent) of $f$, with $\lambda(t)$ scaling its speed. Any reversal in the sign of $\lambda(t)$ would indicate a change in the direction of motion relative to $\nabla f$, corresponding to backtracking—a behaviour that contradicts the optimality of a geodesic. Thus, the monotonic variation of $f(\gamma(t))$ ensures that $\lambda(t)$ does not change sign.

*Equality holds if and only if all $A_i^\gamma(\mathbf{x})$ share the same sign. This implies that the integrand $\frac{\partial f}{\partial x_i}(\gamma(t))\dot{\gamma}_i(t)$ does not change sign for any $i$ or $t$. Consequently, $\nabla f(\gamma(t)) \cdot \dot{\gamma}(t)$ does not change sign, and $\dot{\gamma}(t)$ is everywhere parallel to $\nabla f(\gamma(t))$. That is,*

$$\dot{\gamma}(t) = \lambda(t)\nabla f(\gamma(t)),$$

*for some scalar function $\lambda(t)$.*

*Under the given Riemannian metric $\mathbf{G_x} = \nabla f(\mathbf{x})\nabla f(\mathbf{x})^T$, such paths are precisely the geodesics, connecting $\overline{\mathbf{x}}$ and $\mathbf{x}$.*

***Conclusion:*** *Combining the Necessity and Sufficiency parts above, we can conclude that the equality $\sum_{i=1}^n |A_i^\gamma(\mathbf{x})| = |f(\mathbf{x}) - f(\overline{\mathbf{x}})|$ holds if and only if $\gamma$ is the geodesic path connecting $\overline{\mathbf{x}}$ and $\mathbf{x}$.*

# B    Additional heatmaps and results on Pascal VOC 2012

We also qualitatively compare on Figure 10 Geodesic IG with the original IG on 5 different images of the Pascal VOC 2012 dataset. In these images Geodesic IG heatmaps appears to to have fewer artefacts and is not sensitive to the choice of baseline being a black image. This is contrary to IG, which assigns no importance to the segments of the image that are black, since they have no difference to the chosen baseline.

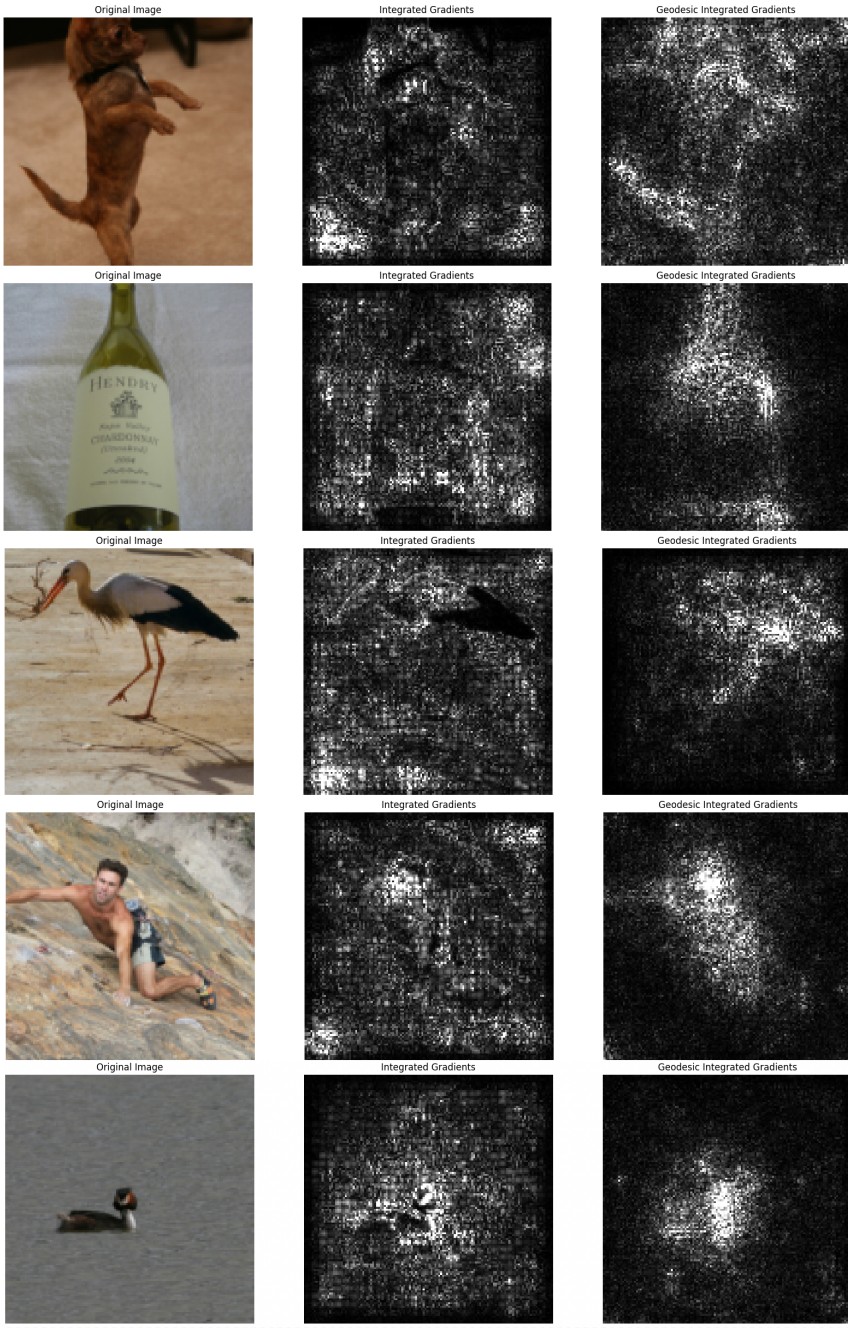

Figure 10: Heatmaps of Integrated Gradients (middle) and Geodesic IG (right) on 5 images from the test set of Pascal VOC 2012.

