# OpenReview forum: "Using the Path of Least Resistance to Explain Deep Networks"
_TMLR — Rejected by TMLR_

### Review · Reviewer_6v6K · 2025-04-01

**Summary Of Contributions:**

This work addresses the unreliability of path-based attribution methods by introducing an alternative approach that computes interpolations on a Riemannian manifold instead of Euclidean space. The authors propose Geodesic Integrated Gradients (GIG), an attribution method that integrates gradients along a geodesic path. To approximate the geodesic path, two techniques are introduced, tailored for models of different sizes. Additionally, the work presents a new attribution axiom, demonstrating that the proposed method exclusively satisfies it. The effectiveness of the proposed method is evaluated through comparisons with other attribution methods on two datasets.

**Audience:**

Yes

**Broader Impact Concerns:**

No need for a broader impact statement.

**Claims And Evidence:**

Yes

**Requested Changes:**

1. The authors are suggested to provide a stronger justification for the proposed axiom, clarifying how it offers meaningful advantages over completeness with cancellation.
2. The authors should reconsider the naming of “Strong Completeness,” as it may mislead readers into thinking it is a strict extension of completeness.
3. The conditions under which each approximation method should be used should be further clarified.
4. The experimental evaluation is expected to be extended by including larger and more diverse datasets to strengthen the empirical evidence.
5. The computational cost introduced by the proposed method should be discussed and compared.

**Strengths And Weaknesses:**

**Strengths**:
1. The paper presents a well-motivated problem, and the idea of integrating gradients in a Riemannian manifold is promising.
2. A new attribution axiom is introduced to account for the cancellation effect, adding a novel perspective to the field.
3. The work explores different approaches to approximate geodesic paths. In addition, theoretical properties are discussed regarding the proposed method, which is an encouraging aspect.

**Weaknesses**:
1. Strong Completeness (SC) axiom lacks the insufficient justification. While the proposed axiom appears reasonable, its advantages over completeness with cancellation are not clearly justified. Unlike the intuitiveness of the standard completeness axiom, the necessity of SC for attribution methods lacks a strong theoretical or empirical foundation. Additionally, satisfying SC does not necessarily imply satisfaction of completeness, making the term “Strong Completeness” potentially misleading. Readers might assume it is a stricter version of completeness when that is not necessarily the case.
2. Geodesic path approximation approaches lack coherence. The two proposed approximation techniques—KNN-based for small models and energy-based for large models—are introduced separately without a clear conceptual connection. It is unclear what constitutes “small” and “large” models in this context. The authors should provide explicit criteria and a justification for selecting one approximation method over the other.
3. Experimental validation is limited and computational analysis is missing. The experiments are conducted on only two small datasets, with real-world evaluations performed on just 100 samples. This is insufficient to demonstrate the method’s effectiveness, particularly given the authors' multiple claims regarding its advantages. The proposed method likely introduces additional computational overhead, but no discussion or comparison of computational time is provided. Understanding the trade-offs in efficiency is crucial for practical adoption.

---

### Review · Reviewer_NFF4 · 2025-04-03

**Summary Of Contributions:**

In this paper, the authors present a new method for determining the importance of input attributes based on axiomatic paths. Previous work (Integrated Gradients (IG)) uses straight paths in an Euclidean space, which the authors consider to be the cause that leads to flawed attributions relevance. Given the above, the authors present Geodesic Integrated Gradients, which replace straight paths with geodesic paths that are defined in a Riemann manifold. To approximate the geodesic path, the authors use K-NN, which is effective but very expensive, which makes them present a new approximation using energy-based sampling. The authors show results on two datasets: synthetic (half-moon) and real (Pascal VOC 2012).

**Audience:**

Yes

**Broader Impact Concerns:**

There are no Broader Impact Statement, and I don’t think they need one.

**Claims And Evidence:**

No

**Requested Changes:**

- Change the structure of the paper to improve its readability. I recommend adding a background section where the authors can explain relevant content (IG, metrics, etc.). With this change, authors can rewrite the introduction and Section 2.
    - Write an introduction that reflects the main ideas of the paper.
    - Consider changing paragraphs where concepts are mentioned and that are explained later in the paper. All content should be explained and then used. This change is not always easy, but it is essential to improve the reading flow.
- Modify the document so there are no sentences with any ambiguity. It is important to think about the reader, who may have other definitions because they come from related but different research backgrounds.
- It is unclear to me how a set of points can be seen as a weighted graph (Page 6). Can the authors add an explanation?
    - This can become critical and more complex when it is mentioned that some graphs can be disconnected.
- Can the authors comment on the following question:
    - When applying these methods in a complex scenario (real-world dataset), multiple possible solutions for the same model may exist. Does this complicate the idea of finding the relevant attributes? Assuming the model is biased, it can focus on multiple unrelated attributes in different solutions. How can these types of methods tackle this issue?
- Can the authors provide a more complete explanation of Axiom 2?
    - Why is it important? Why is a method that fulfills this axiom better? Is there an intuition?
- Increase the title of the figures in Fig 3. Also, maybe normalise the colour in columns (a) and (b) to zero? This change can help to get a better comparison.
- Add the numbers to x and $\hat{x}$ in Fig 5. This is especially relevant because it is mentioned in the caption.

**Strengths And Weaknesses:**

Strengths
- Understanding what models learn is a problem that strongly affects transparency in every Deep Learning model. In this paper, the authors take a step in this direction, proposing a method for finding which attributes are most relevant for the model to classify an input. Although the authors present results and experiments on simple models and datasets, this concept can be extended to other scenarios.
- The authors demonstrate an understanding of the problem, explaining the limitations of previous work and proposing novel solutions. Furthermore, they present limitations of the proposed method, such as the problem of using the kNN-based approximation and its high computational cost.


Weaknesses
- My main concern with the paper lies in its structure and writing. The paper is difficult to read, mainly because of how the authors present and structure each section. Some examples:
    - The second paragraph of the introduction is just a list of papers, without any context or idea.
    - The introduction explains IG in too much detail but does not discuss the motivation of the problem. A background section explaining relevant details from previous works could help with this issue.
    - Figures or concepts are mentioned and explained later in several paper parts. Figure 2 is a great example of this issue since it is presented on Page 2 but explained on Page 12. This issue does not help with the reading flow, as the reader has to look for concepts on other pages constantly.
    - In the phrase: 'IG explains the network', what does it mean to explain in this context? Depending on the reader's background, "explain" can mean different things. Authors need to define or re-write these ambiguities.
    - What does it mean to 'visualize the horizontal attribute'?
- There are comments based on intuition that are unsupported. For example:
    - "... a model that is flat almost everywhere except near the decision boundary, moving slightly further from the decision boundary should not significantly change the model's score." (Page 3) However, this greatly depends on the complexity of the problem. Or are the authors talking specifically about the half-moon problem?
    - "While completeness guarantees certain mathematical consistency …" (Page 3). Can the authors define "Mathematical consistency"?
- Last paragraph of section 2.3. It mentions the importance of finding a suitable value for $\beta$. However, the authors mention in the results the difficulty of finding optimal HP due to the computational cost.
- After Eq 14, the paragraph mentions that purity close to 1 is ideal, and a score of 0.5 reflects random attributes. However, in Table 1, the authors use the AUC-Purity, which completely changes the values, which creates confusion.
    - A similar situation occurs in Table 2, where the authors explain "Comprehensiveness" and "log-odds" (lower is better for log-odds) and then use the AUC that reflects differently (higher is better for log-odds).

---

### Review · Reviewer_ayqH · 2025-04-17

**Summary Of Contributions:**

This paper introduces Geodesic Integrated Gradients (GIG), a novel method for explaining deep learning models, by improving upon the widely used Integrated Gradients (IG). The authors argue that the straight paths used in IG can lead to flawed attributions and introduce geodesic paths as a more reliable alternative. To achieve this, they propose two approximation techniques for computing geodesic paths: one based on k-Nearest Neighbours (kNN) and another based on Stochastic Variational Inference (SVI) for more complex models. The paper also introduces a new axiom, Strong Completeness, and demonstrates that GIG satisfies this axiom, unlike other methods. Empirical results show that GIG outperforms IG and other existing attribution methods in terms of both qualitative and quantitative metrics.

**Audience:**

Yes

**Claims And Evidence:**

Yes

**Requested Changes:**

1. Explore more scalable solutions for large datasets and investigate the trade-offs between accuracy and computational efficiency in the context of large models.
2. Analyze the impact of hyperparameters on the model’s performance to provide clearer guidelines for future implementations.
3. Include comparisons with more attribution methods beyond IG (e.g., SHAP, LIME).

**Strengths And Weaknesses:**

Strengths:

The idea of using geodesic paths instead of straight lines for model attribution is an insightful contribution. The introduction of the Strong Completeness axiom provides a theoretical guarantee that is novel and valuable for improving the rigor of attribution methods. The experiments demonstrate the practical effectiveness of the proposed method in a variety of contexts.

Weaknesses:

1. While the authors acknowledge the high computational cost of the proposed method, particularly with SVI, the paper does not fully explore potential strategies for improving efficiency. The current reliance on SVI and kNN may limit the scalability of GIG.
2. While the paper addresses the limitations of IG, there is limited discussion on the potential shortcomings or limitations of GIG itself. The issue of graph disconnections in the kNN approximation is only briefly mentioned, but a more detailed discussion on the limitations of the kNN method in highly complex spaces could be useful.
3. The authors mention the challenge of selecting appropriate values for parameters in the energy-based sampling method. However, there is not enough exploration on how these parameters might affect the performance of GIG.
4. While the paper compares GIG with IG and other attribution methods, the comparison could be more comprehensive. For example, a more detailed analysis of how GIG compares with other recent attribution methods like SHAP or LIME would be helpful.

---

### Public Comment · ~Eslam_Zaher1 · 2025-05-17
**The paper proposes Geodesic Integrated Gradients (GIG), which closely overlaps with our earlier work on Manifold Integrated Gradients (MIG), published at ICML 2024. Both methods integrate gradients along geodesics on a Riemannian manifold to address limitations of Integrated Gradients. Despite strong conceptual and methodological similarities, our work was not cited. We believe this is a significant oversight that should be acknowledged and corrected.**

This paper proposes Geodesic Integrated Gradients (GIG) as a path-based feature attribution method that integrates gradients along geodesic curves defined on a Riemannian manifold. While this is a valuable and well-presented contribution, I was surprised to see no mention of our earlier work: Zaher et al. (2024), “Manifold Integrated Gradients: Riemannian Geometry for Feature Attribution”, which was published at ICML 2024 and is available at arXiv:2405.09800.

Our work introduces Manifold Integrated Gradients (MIG), which also integrates gradients along geodesics on a Riemannian manifold, motivated by the same fundamental limitations of Euclidean straight-line paths in Integrated Gradients. The similarity in the motivation, theoretical underpinnings (e.g., Riemannian metrics induced by the model or generator), and methodology (e.g., use of geodesics for attribution) is nontrivial. We also cited and built upon the same foundational literature, including the axiomatic framework of IG and geometric criticisms of linear paths.

While differences exist (e.g., our use of VAEs to define the data manifold vs. their kNN and variational methods), the conceptual and methodological overlap is significant. Given that our work has been publicly available and addresses nearly identical goals using very similar tools, I believe it should have been cited and discussed as related work. The absence of citation—particularly since the current paper was submitted well after ours was publicly posted—risks presenting a misleading sense of novelty to readers unfamiliar with the field.

I hope the authors can acknowledge this oversight and clarify the relationship between GIG and MIG in any revised version.

---

> ### Author Response · Authors · 2026-02-05
>
> Thank you for bringing Zaher et al. (2024) to our attention and sincerely apologise for this oversight. MIG is excellent work that we should have cited, and we will do so prominently in any future revision.
> To provide context: an earlier version of this work was submitted to ICML 2023 (https://openreview.net/forum?id=c8iDBe2YhX), before MIG was publicly available, and our core ideas were developed independently during that period. When preparing this submission, we failed to conduct a sufficiently thorough literature review, which led to this omission. We appreciate the commenter's understanding that this was an oversight rather than intentional.
> We believe GIG and MIG represent complementary approaches to a shared insight, that geodesic paths offer advantages over straight-line interpolation for feature attribution. MIG elegantly uses a VAE to learn the data manifold and derives its Riemannian metric from the decoder's Jacobian, ensuring paths remain within the data distribution. GIG takes a different approach by deriving the metric directly from the explained model's Jacobian, avoiding the need for an auxiliary generative model but with different geometric properties. We see these as exploring different points in a design space rather than one superseding the other, and we look forward to discussing their relative merits in the revised manuscript.
> We will update the related work section to properly cite and discuss MIG, clarify the relationship between our approaches, and ensure readers can appreciate both contributions. Thank you again for raising this important point.

---

### Decision · Action_Editor_GdFN · 2025-05-23

**Recommendation:** Reject

**Comment:**

Reviewers raised many concerns about unsupported statements, insufficient empirical evaluations, etc., and gave many requests, on which reviewers provided no feedback.

A public comment raised another serious concern, pointing out the existence of prior work, with which the contributions of the current submission has significant overlap.

Zaher et al. (2024), “Manifold Integrated Gradients: Riemannian Geometry for Feature Attribution”, which was published at ICML 2024 and is available at arXiv:2405.09800.

Accordingly, the paper requires major revision, considering the prior work as a baseline, and highlighting the difference and advantages of the propose methods, as well as reflecting all reviewers' comments.

**Audience:**

Yes.

**Claims And Evidence:**

No.  Reviewers pointed out some claims are not supported.